# Pharmacoepidemiological Evaluation in Prostate Cancer—Common Pitfalls and How to Avoid Them

**DOI:** 10.3390/cancers13040696

**Published:** 2021-02-09

**Authors:** Aino Siltari, Anssi Auvinen, Teemu J. Murtola

**Affiliations:** 1Faculty of Medicine and Health Technology, Tampere University, 33520 Tampere, Finland; aino.siltari@helsinki.fi; 2Department of Pharmacology, Faculty of Medicine, University of Helsinki, 00014 Helsinki, Finland; 3Faculty of Social Sciences, Tampere University, 33014 Tampere, Finland; anssi.auvinen@tuni.fi; 4Department of Urology, TAYS Cancer Center, 33520 Tampere, Finland

**Keywords:** pharmacoepidemiology, prostate cancer, metabolism, common biases, confounding, retrospective studies

## Abstract

**Simple Summary:**

Pharmacoepidemiologic research provides opportunities to evaluate how commonly used drug groups, such as cholesterol-lowering drugs, may affect the prostate cancer risk or mortality. However, such studies need to be carefully designed in order to avoid biases caused by systematic differences between medication users and non-users. Similarly, data must be carefully analyzed and interpreted while acknowledging possible biases that can lead to erroneous conclusions. Here, we review common pitfalls in such studies and describe ways to avoid them in an effort to aid future research.

**Abstract:**

Pharmacoepidemiologic research provides opportunities to evaluate how commonly used drug groups, such as cholesterol-lowering or antidiabetic drugs, may affect the prostate cancer risk or mortality. This type of research is valuable in estimating real-life drug effects. Nonetheless, pharmacoepidemiological studies are prone to multiple sources of bias that mainly arise from systematic differences between medication users and non-users. If these are not appreciated and properly controlled for, there is a risk of obtaining biased results and reaching erroneous conclusions. Therefore, in order to improve the quality of future research, we describe common biases in pharmacoepidemiological studies, particularly in the context of prostate cancer research. We also list common ways to mitigate these biases and to estimate causality between medication use and cancer outcomes.

## 1. Introduction

The widely accepted theory of carcinogenesis is that it involves a sequential development; first, a local primary tumor develops, progressing over time to adopt a more aggressive phenotype, finally invading other tissues and forming metastases. The hallmarks of the progression of cancer to metastatic disease includes the cancer cells’ ability to sustain proliferative signals and evade growth suppressors, resist cell death, induce angiogenesis, enable replicative immortality, and finally develop an ability to invade the surrounding tissues and form metastases [1]. Genetic instability, changes in energy metabolism, the ability to evade immune defense, and boundless growth have key roles in every step of carcinogenesis, leading to the creation of a microenvironment that favors tumor development and progression. The hallmarks of cancer have been comprehensively reviewed by Hanahan and Weinberg (2011) [1].

Many of the typical adaptation properties utilized by cancer cells, especially those affecting energy metabolism, can be targeted with established drugs currently used for other indications, e.g., the antidiabetic drug metformin and the cholesterol-lowering statins. Potential cancer preventive effects of such established drugs can be evaluated in several ways—in vitro studies are helpful in clarifying cancer-preventive mechanisms in cancer cell lines whereas randomized clinical trials are needed to evaluate a drug’s clinical efficacy. Pharmacoepidemiological studies provide an important opportunity to evaluate how established drug groups may associate with the cancer risk and outcomes in large study populations, providing important real-life data on possible drug efficacy.

In this review, we performed a pharmacoepidemiological evaluation of potential cancer-preventive drugs in prostate cancer (PrCa). We reviewed the most common pitfalls encountered in this type of research, especially when studying cancer-related events, and present ways to avoid these obstacles so that the research results will be as reliable and free of bias as possible.

## 2. Biological Rationale for Pharmacoepidemiological Studies in Prostate Cancer

### 2.1. Energy Metabolism

In many cancer types, the energy metabolism of the tumor cells becomes changed to favor rapid cell growth and proliferation [1]. A common shift is termed the Warburg effect; cancer cells avoid oxidative phosphorylation and favor anaerobic glycolysis even when oxygen is available [2,3]. Glycolysis produces less energy than oxidative phosphorylation but produces material for macromolecules, thus enhancing carcinogenesis [4,5]. Glycolysis produces lactate that is converted by tumor stromal cells into pyruvate, which is utilized as an energy source [6,7]. Tumor cells often have an increased uptake of glucose due to the upregulated expression of glucose transporters on the cell surface [8]. This increased glucose flux and metabolism are clinically used in positron emission tomography imaging of tumor cells via the accumulation of a radiolabeled glucose analog in the primary tumor tissue and metastases [9]. The Warburg phenomenon is not commonly encountered in prostate cancer, but evidence for the importance of glucose metabolism exists also in this cancer type [10].

The role of hyperinsulinemia, insulin receptors (IR), and insulin-like growth factor (IGF) on cancer development and progression have attracted scientific interest due to the global epidemics of obesity and type 2 diabetes [11]. In prostate cancer, the overexpression of both IR and IGF has been shown to present in around 10–30% of patients, and IGF in particular has been linked to advanced prostate cancer [12,13]. Furthermore, androgen deprivation therapy (ADT), a standard treatment for advanced PrCa, causes hyperinsulinemia as a side effect of treatment [14], which might partly represent a mechanism for the progression of hormone-independent PrCa. In cell models of PrCa, hyperinsulinemia has been shown to increase cellular proliferation, invasiveness, and the activation of cellular plasticity mechanisms [15,16].

In preclinical studies, the use of the antidiabetic drug metformin has been postulated to affect prostate cancer development and progression in multiple ways: through AMPK-dependent and -independent mechanisms, changing activity in the IGF-1 signaling pathway, suppressing the androgen receptor pathway, inhibiting the mTOR pathway, as well as impacting on lipogenesis [17]. However, according to a meta-analysis of epidemiological studies, the use of metformin has not been associated with either the prostate cancer risk [18] or the disease-specific mortality [19].

### 2.2. Cholesterol Metabolism

Cholesterol is an important compound of cellular membranes and substrate for many biological compounds such as steroid hormones including androgens. Serum cholesterol is produced either by de novo synthesis through the mevalonate pathway or by uptake of dietary cholesterol.

Cholesterol metabolism is important to allow the growth of cancer cells, but it has also been linked to cell proliferation, migration, and invasion [20,21]. Cholesterol is the precursor for biosynthesis of all steroid hormones, including androgens. Therefore, it is logical to assume cholesterol metabolism to be important, especially in hormone-dependent cancer types. Indeed, upregulation of cholesterol and lipid biosynthesis is a known hallmark of prostate cancer. In advanced prostate cancer, the de novo synthesis of cholesterol is increased, while the expression of the low-density lipoprotein (LDL) receptors and activity in cholesterol esterification pathways are downregulated [22]. Thus, prostate cancer cells rely more on de novo synthesis of cholesterol rather than dietary cholesterol. Further, esterified cholesterol can be stored and accumulated in lipid droplets in high-grade and metastatic prostate cancer cells. In a mouse model, it was demonstrated that a deletion of these droplets inhibited cancer cell proliferation, as well as the invasion capability of the malignant cells, thus preventing tumor growth [23].

In epidemiological studies, hypercholesterolemia has been considered as a possible risk factor for prostate cancer progression—the risk of disease recurrence after primary treatment is significantly elevated in men with hypercholesterolemia compared to men with normal serum cholesterol levels [24].

Cholesterol-lowering drugs, i.e., statins, inactivate the mevalonate pathway by inhibiting 3-hydroxy-3-methyl-glutaryl-coenzyme A (HMG-CoA) reductase. In epidemiological studies, the use of statins has been associated with a lowered risk of prostate cancer recurrence and progression [25,26], but not with the overall risk of prostate cancer diagnosis [27,28]. In a randomized clinical trial, an intervention with atorvastatin lowered the concentration of serum prostate-specific antigen (PSA) as compared to placebo in men with high-grade prostate cancer [29]. Treatment also decreased tumor proliferation after a minimum exposure of 27 days. In another trial, treatment with fluvastatin before prostatectomy showed promising effects on tumor cell apoptosis [30]. Statin treatment is also associated with better survival in men with advanced prostate cancer managed with androgen deprivation therapy; statin use has been associated with an 8–10 month longer response to androgen deprivation when compared to non-users [31].

### 2.3. Metabolic Adaptions to Hypoxic Tumor Microenvironment

Hypoxia is common during cancer progression due to the rapid growth of tumors. Hypoxia is consistently associated with resistance to oncological treatments [32,33]. It is believed that metabolic alterations in cancer cells likely contribute to the cells’ ability to withstand hypoxic conditions. Reliance on glycolysis demands that the cells have access to an energy source and materials for proliferation despite hypoxia. In fact, a hypoxic microenvironment further induces glycolysis by upregulating glucose transport and the enzymes needed for glycolysis [34]. Sterol regulatory element-binding proteins (SREBPs) control cholesterol metabolisms and are activated by hypoxia [35,36]. SREBPs are transcription factors that upregulate the genes involved in the cholesterol pathways [37,38]. Hypoxia induces the expression of hypoxia-inducible factors (HIFs) that regulate many hypoxia-related metabolic changes. Thus, the presence of a hypoxic environment also regulates lipid metabolism in a HIF-dependent manner [39]. In prostate cancer cells, aberrant lipid metabolism has an important role in overcoming the hypoxic microenvironment [40].

### 2.4. Other Target Mechanisms for Evaluation in Pharmacoepidemiological Studies

Inflammation, especially chronic inflammation, plays a key role in the development and progression of many cancers. There is evidence that chronic inflammation also plays a role in PrCa etiology (for review, see [41]). The commonly used analgesics non-steroidal anti-inflammatory drugs (NSAIDs) reduce inflammation, and thus the use of these drugs, especially acetylsalicylic acid (aspirin), has often been a focus of pharmacoepidemiological studies on prostate cancer. A meta-analysis evaluating 29 studies concluded that use of aspirin was associated with a slightly (approximately 10%) decreased risk of PrCa; however, the overall use of all NSAIDs was associated with a slightly increased risk [42]. Subsequently, a meta-analysis of 39 studies concluded that use of aspirin and NSAIDs was associated with a lowered risk of PrCa-specific death [43]. We have also reported similar results among the Finnish population [44,45].

Other drugs that have been pharmacoepidemiological targets for prostate cancer development and progression are antihypertensive drugs as well as allopurinol and anticoagulants. Compounds inhibiting the renin–angiotensin–aldosterone system and beta-blockers have been the most widely investigated antihypertensive drugs [46,47]. Platelets have been suggested to play a role in promoting tumor metastasis, but the association between anticoagulant therapy and prostate cancer risk is controversial [48,49]. In addition, it has been reported that treatment with anticoagulants may be associated with an increased risk of cancer death [50]. Allopurinol may possess anti-inflammatory properties [51]; in a pharmacoepidemiological study, its use was associated with a reduced risk of prostate cancer [52].

## 3. Commonly Used Data Sources in Pharmacoepidemiological Studies

### 3.1. Registries

Many countries have registries that routinely collect healthcare data either nationally, such as the UK National Health Service, or from a clearly outlined population, such as members of a certain healthcare plan, such as Kaiser Permanente in the USA. Data is collected routinely either continuously or between certain time intervals. In the Nordic countries, national registries have been shown to be accurate, thus representing a valuable resource for pharmacoepidemiological studies. In addition, data from different registries can be combined easily and reliably using personal identification numbers [53]. Many different routinely collected registries are available, such as national cancer registries, drug prescription databases, the causes of death registries, and hospital outpatient registries.

### 3.2. Surveys

When appropriate routinely collected registries are not available, data can be collected using surveys. Information on medication use is collected at one or more time-points. Since the surveys are filled in by the participants, the information may not be as accurate and reliable as in registries with routinely collected data, especially if survey data are collected retrospectively at the time of cancer diagnosis or after it; in other words, the quality of survey data may systematically differ between cancer patients and controls. When survey data includes personal identifiers compatible with registries, such as a social security number, survey data can be combined with registry data. Optimally these two data sources can be complementary; for example, comprehensive registry data on medication use can be complemented with survey data on lifestyle factors such as smoking and diet. There are several examples of successful studies based on survey data, e.g., the Physician’s health study [54] and the FINRISK study [55]. For instance, FINRISK is a large Finnish population survey on risk factors for chronic, non-communicable diseases that has been carried out every five years since 1972. These survey data have been combined with Finnish national registries to evaluate the risk for various health outcomes.

## 4. Common Pitfalls in Pharmacoepidemiological Research

### 4.1. Immortal Time Bias

In general, the immortal time bias refers to a situation occurring when one comparison group has periods of follow-up when the study outcome cannot occur (Figure 1). In pharmacoepidemiology, this typically occurs when cohort follow-up starts at the baseline time point, e.g., cancer diagnosis, but medication use could have started at any time point after the baseline. Therefore, follow-up time before the initiation of medication use is termed “immortal time” as the users lived to subsequently become medication users. This bias favors medication users, especially in observational cohort studies assessing mortality according to medication use. Thus, when a flawed approach is used in the design and/or in the data analysis of the results, it might lead to immortal time bias, which can generate an illusion of treatment effectiveness [56]. The immortal time bias is a problem, especially in pharmacoepidemiological studies that compare users of a certain drug or drug group against non-users [57].

Lévesque et al. (2010) [58] listed criteria for identifying immortal time bias in cohort studies: (1) is the treatment status determined after the start of follow-up or defined using follow-up time?; (2) is the start of follow-up different for the treated and untreated group relative to the date of diagnosis?; (3) have the treatment groups been identified hierarchically?; (4) are subjects excluded on the basis of treatment identified during follow-up?; and (5) has a time-fixed analysis been used? If a study does not take into account the above-mentioned criteria, then an immortal time bias may occur. The use of the time-dependent variable is especially recommended as a way of avoiding the immortal time bias [59].

The immortal time bias can be controlled with time-dependent exposure variables in which exposure status is updated during the follow-up. In pharmacoepidemiology, this means that medication usage status changes, being a non-user before the first documented drug purchase or report of use and changing to becoming a user after that event. This eliminates follow-up time where a user would be falsely categorized as exposed before the actual start of usage. We used time-dependent variables and the start of follow-up from the beginning of exposure when the impact of commonly used drugs on prostate cancer risk or mortality is compared against that in non-users [46,60]. Use of these methods requires, at minimum, knowledge on starting dates of medication use. Future studies should make every possible effort to obtain these data in order to control for immortal time bias.

### 4.2. Time-Window Bias

Suissa et al. (2011) [61] described a time-window bias in case–control studies demonstrating that the protective impact of statin use on lung cancer was due to the longer time-window for measuring exposure in controls rather than in cases; among the cases, the statin exposure was limited to usage that occurred before lung cancer diagnosis, whereas in controls, no similar limitation was applied, and statin use could have occurred over a longer time period. Therefore, controls had a greater likelihood of being statin users, creating a bias that lowered the risk association between lung cancer and statin use (Figure 2). This time-window bias can be avoided by assessing equal time windows to observe exposure in cases and their matched controls. Usually this is done by using the diagnosis date of the case in a matched case–control pair as an index date also for the control. Medication use is limited to occur before the index date both for the case and the control, ensuring equal exposure times. Suissa et al. (2007) [56] proposed the application of time-dependent sampling, i.e., ensuring equal exposure time to cases and controls to avoid the problem of the time-window bias. Di Martino et al. (2015) [62] demonstrated how results change markedly in the same study population due to time-window bias, i.e., differences when time-dependent sampling was used compared to time-independent sampling.

### 4.3. Bias Caused by Selective Discontinuation of Drugs at Terminal Phase of Cancer

When cancer has progressed to the terminal phase of cancer progression, it is common clinical practice to discontinue all drugs that have no direct palliative impact on pain relief or other symptoms. This commonly includes preventive drugs such as statins, antihypertensive drugs, and oral antidiabetic drugs [63]. The practice can create a powerful bias in pharmacoepidemiological studies on cancer mortality if medication use during the final follow-up year affects the exposure status. The bias creates an illusion that non-users, who might have been users for a long time but discontinued usage shortly before cancer death, have more cancer deaths compared to those who have kept using the drug until the end of follow-up (Figure 3). Our approach to avoid this bias while analyzing medication use as a time-dependent exposure is to keep the subjects as users after the first recorded usage, i.e., user status may change from non-user to user but will not change back to non-user even though the medication use has been later discontinued [46]. This effectively eliminates the selective discontinuation bias described above. A limitation of this approach is that it overestimates the length of medication use in those who had used medication only for a very short period of time. This limitation can be overcome by using washout periods, where medication users with only short exposure periods are excluded.

### 4.4. Confounding by Indication

Confounding by indication is a common source of bias in observational pharmacoepidemiological studies as treatment allocation is not randomized. Therefore, medication users differ systematically from non-users, at least in the condition for which the medication was prescribed. Especially in the context of cholesterol-lowering and antidiabetic drugs, multiple comorbidities associate with medication use and with cancer outcomes, providing a further potential source of bias. Confounding by indication can either overestimate or underestimate the risk association with the outcome depending on the association between the underlying condition requiring medication use and the cancer [64].

One example of this situation occurred when we investigated the association between use of antihypertensive drugs and the risk of prostate cancer death in a Finnish population-based cohort [46]. In the preliminary analysis, the use of diuretics was associated an increased risk of prostate cancer death compared to non-users (hazard ratio 2.61, 95% CI 2.22–3.06). Furthermore, the risk association was strongest in men who used drugs at a low dose and only for a short time (1 year or less) prior to the end of follow-up (hazard ratio 3.17, 95% CI 2.6–3.87). When the analysis on diuretics was broken down to allow a separate analysis for each drug in the diuretics class, we found that the increased risk was associated with two compounds in particular, namely, furosemide and spironolactone. These drugs are commonly used in the treatment of oedema and fluid retention, both of which are common problems in advanced, terminal phase cancer, prostate cancer included. Thus, the increased risk for PrCa death among diuretic users was due to confounding by indication; more prostate cancer deaths occurred in diuretic users than non-users because these drugs were actually being used to manage the complications of advanced prostate cancer. Thus, when the use of loop-diuretics and spironolactone was excluded from the final analysis, we estimated that the hazard ratio decreased to 1.25 (1.05–1.49).

One option to avoid such a bias is to use a washout period at the beginning of the exposure; confounding by indication is likely to have the greatest impact at the beginning of medication use, as demonstrated by our example of prostate cancer mortality among diuretics users. Therefore, exclusion of a reasonable time period from the beginning of medication use can help to mitigate this bias. Another very beneficial way to evaluate confounding by indication is to include in the analysis all drugs that have the same indication of use but differing mechanisms of action. To avoid biases due to confounding by indication, one calculates the risk estimates for all drugs used for that indication, independent of the mechanism of action. If, however, one drug group would exhibit a different association with cancer outcomes compared to other drugs used for same indication, this would suggest that the mechanism of that particular drug group may be exerting an oncological impact. For example, if antihypertensive drugs affecting the renin–angiotensin–aldosterone (RAA) system were associated with a lowered risk of prostate cancer death, whereas other antihypertensive drug groups were associated with no decrease in risk or even an increased risk, this would suggest a protective effect for RAA inhibition [46].

### 4.5. Protopathic Bias

A protopathic bias is another common source of bias that should be taken into consideration when designing and interpreting pharmacoepidemiological studies. In cancer epidemiology, a protopathic bias occurs when the drug of interest is used to treat symptoms caused by a still undiagnosed cancer, i.e., drugs have been prescribed for an undiagnosed or pre-stage of the disease [65]. An excellent example of protopathic bias occurs when the association between analgesic drugs and risk of advanced prostate cancer or cancer mortality is being investigated. Such studies find a strong association between analgesic medication use and the risk of advanced prostate cancer, especially for short-term medication use. In this case, the risk association is not caused by analgesic drugs, but by treatment of metastasis pain, a symptom that often precedes the diagnosis of advanced prostate cancer, as prostate cancer predominantly metastasizes to bone [44,45]. It is important to take this bias into account when studying the association between medication use and slowly progressing and potentially symptomatic diseases such as cancer. Several options to control for protopathic bias are available. One option is to exclude medication use from the time assumed to be affected by the possible protopathic bias, e.g., in the case of analgesics, this would mean an exclusion of the first year of medication use as this would considerably limit any protopathic bias in the association with the risk of advanced prostate cancer.

Another useful option to control and evaluate the protopathic bias is to adopt lag-time analyses (Figure 4) [66]. In this approach, the exposure is lagged with a follow-up time, e.g., with a one-year time lag, the cancer risk is not evaluated by medication use that occurred at the time of diagnosis, but by usage that occurred one year before that diagnosis. Thus, the time period immediately preceding cancer diagnosis is excluded, avoiding, or at least mitigating, the protopathic bias. We have also used this concept in our analysis [60].

### 4.6. Recall Bias

In pharmacoepidemiological studies, the risk of recall bias is evident especially in studies where surveys are used to collect data on past medical use. The typical setting where this bias is encountered is a case–control study where cases are cancer patients treated in an outpatient clinic; the controls are patients without cancer but treated in the same outpatient clinic or hospital for other indications. A recall bias arises when patients who have become diagnosed with a potentially life-threatening disease will fill in surveys more comprehensively or in greater detail as compared to patients with a less serious condition. Patients who have fallen seriously ill often do a lot of soul-searching and think about what could have caused the disease. The compliance of these patients to fill in survey questionnaires is often higher and more accurate than that of less seriously ill patients. This may lead to an overestimation of the association between the drug of interest, length of use, and outcome in the cases. To avoid this bias, routinely collected data, e.g., hospital or national registries are a preferable data source, as data availability does not depend on the patient’s conditions or memories. In addition, the control group should be selected cautiously so that it resembles the patient group as much as possible. One way to assess this kind of bias is to investigate whether the compliance in filling of surveys is similar between the cases and controls.

### 4.7. Healthy User Bias

The healthy user bias is a form of selection bias that occurs in pharmacoepidemiology when medication users generally follow a healthier lifestyle than non-users. This situation occurs especially when drugs that affect a non-symptomatic risk factor, such as cholesterol-lowering or antihypertensive drugs, are used in primary prevention, i.e., to lower the risk for an adverse health outcome. The willingness to take primary preventive medications is a signal of compliance with medical advice and health-seeking behavior. Medication users generally have a healthier lifestyle also in other areas, e.g., having a healthy diet and regular exercise habits and more active participation in cancer screening programs. Prostate cancer incidence and mortality in a given population are influenced by frequency of PSA testing; therefore, in the context of prostate cancer, healthy user bias may cause falsely elevated overall prostate cancer risk but lowered disease-specific mortality among users compared to non-users due to more active participation in PSA testing.

Subjects who are using prescription drugs for primary prevention also meet physicians on a regular basis. Thus, their health status is monitored more actively than non-users. Therefore, statin users who use the drugs for primary prevention of coronary artery disease are prone to this bias; such statin users have a decreased risk of various health outcomes unrelated to effects of statins, such as a lowered risk of motor vehicle accidents and workplace accidents [67]. On the other hand, statins are more commonly used for secondary prevention, i.e., to prevent worsening of some established cardiovascular disease. In the secondary prevention group, the healthy user bias is reversed; patients with cardiovascular disease have more risk factors, such as smoking, in comparison to non-users. Therefore, when considering the possibility of a healthy user bias, it is important to consider whether the drugs have been used for primary or secondary prevention.

### 4.8. Bias Due to Competing Causes of Death

This source of bias occurs in pharmacoepidemiology when medication users have an elevated risk of dying of non-cancer causes. For example, users of cholesterol-lowering, antihypertensive, and antidiabetic drugs have cardiovascular risk factors that indicate the medication use. Therefore, they are at an increased risk of cardiovascular morbidity and mortality compared to non-users and thus their increased cardiovascular mortality may introduce a bias, lowering the observed risk estimates for cancer mortality, as these medication users do not live long enough to die later of cancer. This may be especially relevant in prostate cancer, where most patients are elderly men with often multiple comorbidities. This bias can be estimated by analyzing separately the risk for the presumed competing cause. One common way to control this bias is to use a competing risks regression model in the analysis, such as that described by Fine and Gray (1999) [68].

## 5. Tools to Assess Causality and Control Bias

### 5.1. Assessment of Temporal and Dose Dependence Between Medication Use and Cancer Outcomes

The rules of causality include that there should be a temporal and quantitative relationship between the exposure and the outcome. Pharmocoepidemiological studies should always aim to estimate whether the risk association between cancer and the studied drug changes with the duration or dose of medication use. In a causal association, there should be a correlation between changes in the outcome risk with the duration and amount of the exposure, becoming stronger in conjunction with long-term/high-dose use. On the other hand, the selection bias related to the beginning of drug use, such as protopathic bias and confounding by indication, mainly affects short-term use, as we presented above in our example of the use of diuretics when prostate cancer-specific mortality was evaluated [46]. In the long-term use of drugs, the impact of selection bias tends to even out over time, and thus the strongest association is seen among the short-term users. Therefore, if the dose and time dependency cannot be analyzed, the causal association between drug exposure and the outcome cannot be comprehensively evaluated.

### 5.2. Assessment of Adherence to Medication Use

The assessment of adherence to medication use is particularly a problem when data are collected from prescription registries collecting information on prescriptions or drug purchases; purchase data are available, but information on whether the subject actually used the drug is missing. Therefore, medication use may appear to be greater than what actually happened. This can create a bias by diluting the observed risk associations according to medication use. On the other hand, in studies where information on medication use is collected with surveys, subjects might exaggerate or underestimate their actual usage. In the absence of documented medication intake, there is no sure way to evaluate or mitigate this bias. It is logical to assume that it would affect mostly short-term use, whereas those who have purchased the drugs repetitively over a long time span can be assumed to have actually consumed the drugs. If measurements of blood glucose or cholesterol levels are available, compliance with antidiabetic and cholesterol-lowering medication use can be evaluated indirectly by observing changes in these parameters.

### 5.3. Propensity Score and Instrumental Variables

In order to obtain reliable results from retrospective observational studies, we must not only control for factors causing the above-mentioned biases, but also we must adjust for potential confounding factors. When the data include information on such confounding factors, adjustment is commonly done by adding these as variables into the regression models used to calculate the relative risks for the outcome. In small datasets in particular, adjustment for multiple confounding factors can be problematic, as confidence intervals for the risk estimates tend to become wider along with the number of model adjustments. One remedy for the problem is to use adjustment for propensity score instead of multiple variables [69]. In this method, the association is evaluated between the potential confounding factors and the exposure. For example, odds ratios for statin use by smoking, obesity, antihypertensive drug use, and antidiabetic drug use can be calculated. On the basis of these odds ratios, we can calculate an individual propensity score for statin use for each person in the study population in terms of how many of these risk factors they have. The propensity score can be used in multiple ways: (1) model adjustment—adjusting regression model for propensity score allows for an adjustment for multiple potential confounding factors in one variable, reducing the toll on statistical precision; (2) matching—case–control pairs can be matched according to their propensity scores, creating a situation where the case and the control are equally likely to be medication users in terms of their known background characteristics; and (3) stratification—stratifying the data by propensity score divides the study population into groups homogenous in their likelihood to be medication users.

Thus, the propensity score is a method that can be used to balance known background baseline variables between exposed and non-exposed study subjects or cases and controls. There are several different methods available for calculating propensity scores [70].

One limitation of the propensity score method is that it does not mitigate systematic biases or confounding by unknown background variables. The propensity score has the greatest value in studies with small study populations, whereas in large datasets with tens of thousands of subjects, the use of a propensity score does not confer any additional benefit over separate adjustment for each background variable [71].

The instrumental variable method can be used to address unmeasured confounding [72]. This means using a variable that is related to the exposure but not with the outcome as a way to assess causality. In cancer epidemiology, a common application is to evaluate the genotype known to predict a certain trait, such as hypertension to assess the causality between hypertension and cancer. In pharmacoepidemiology, this may give opportunity for indirect estimations, as genotype may predict a trait that indicates medication use but does not mean that medication has actually been used. The use of this method is recommended only when unmeasured confounding is a major concern as the adoption of this method lowers statistical power of the risk estimates, e.g., creating high confidence intervals.

### 5.4. Other Study Setups to Control Bias

Lund et al. (2015) [73] reviewed the concept of the active comparator, a new user study design in pharmacoepidemiological research. This study design aims to mimic the design of a randomized controlled trial. In this design approach, the study population is limited to participants with documented indication to use the medication of interest, e.g., elevated blood glucose concentrations. Additionally, the participants must have no documented medication usage before the baseline. This active comparator part helps to restrict the study to subjects with an indication for treatment without contraindications. Then participants who start using the drug of interest, such as metformin, after baseline are compared to those who start using another drug such as insulin. Including only usage post-baseline ensures correct timing between the covariate and the exposure, and thus the adoption of this design helps to avoid problems caused by confounding by indication and some forms of selection bias. The exclusion of participants with medication use before the baseline makes an evaluation of the exposure more accurate, as medication use data before the baseline are often incomplete, whereas this is not the case for usage data during the study period.

### 5.5. Further Considerations

A pharmacoepidemiological evaluation is always retrospective and the ability to adjust for the above-mentioned confounding factors and biases evidently depends on the quality of the available data; furthermore, it is commonly not possible to control for all of these factors. Still, every effort should be made to ensure that the data have as high a quality as possible. Often this means combining data from many different registers with survey data. The above-mentioned biases should already be taken into account in the study’s design, and the data need to be carefully analyzed and interpreted while acknowledging the possibility of biases if the data do not allow for their control.

It is also important to systematically assess the quality of the data sources; completeness and validity of the register data should be evaluated as should the response activity in a survey. There may well be validation studies available for some widely used registries. For instance, Sund (2012) [74] systematically reviewed the quality of the Finnish hospital discharge register and concluded that data completeness and accuracy varied from satisfactory to very good as long as the recognized limitations were taken into account.

## 6. Conclusions

In conclusion, we describe the biological rationale behind a pharmacoepidemiological study on commonly used medications and their possible association with the development and progression of prostate cancer. Such studies hold great possibilities to enhance understanding of prostate cancer etiology and identify new targets for anticancer interventions. Nevertheless, such studies are prone to multiple sources of bias, which need to be acknowledged and addressed by all pharmacoepidemiological studies to avoid making erroneous conclusions, which would lead to wasting of research resources to study a risk association that was biased in the first place. We have highlighted several sources of bias and recommend methods to avoid these commonly occurring biases in pharmacoepidemiological studies aimed at identifying associations between medication use and cancer risk or outcomes. When designing a study, the completeness and validity of the register data should be evaluated before conducting any analysis. In the analysis, it is essential to control for an immortal time bias and a time-window bias. Finally, when inferring causality of the risk association, we must evaluate the time and dose dependency between medication use and the outcome. A lag-time analysis can help to control for a protopathic bias, and the adoption of the propensity score method can help in controlling for confounding by known background variables. If data are available, other drugs with the same indication but different mechanisms of action should be evaluated to estimate confounding by indication. This approach is essential when evaluating a drug or drug group with a given mechanism of action, such as those affecting cancer metabolism. When these biases have been properly addressed, pharmacoepidemiological studies provide invaluable source of real-life information on how commonly used drugs may affect prostate cancer.

## Figures and Tables

**Figure 1 cancers-13-00696-f001:**
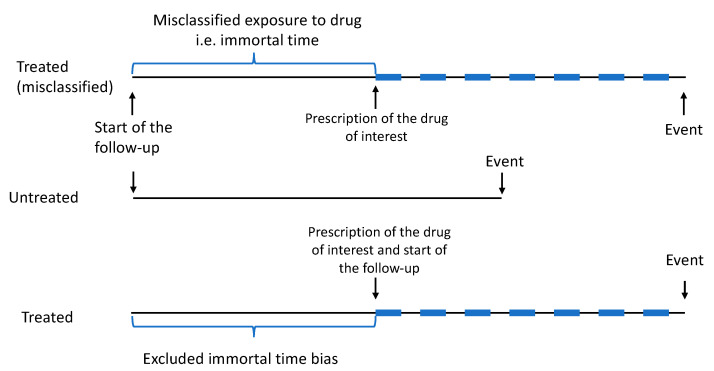
Illustration of time immortal bias in an observational cohort study (treated (misclassified)) and one solution to avoid it (treated). Immortal time bias can generate an illusion of treatment effectiveness when it does not actually occur. This bias can be avoided by using time-dependent variables where exposure status is updated during the follow-up, or alternatively by excluding all of the non-exposed follow-up times from the beginning of exposure.

**Figure 2 cancers-13-00696-f002:**
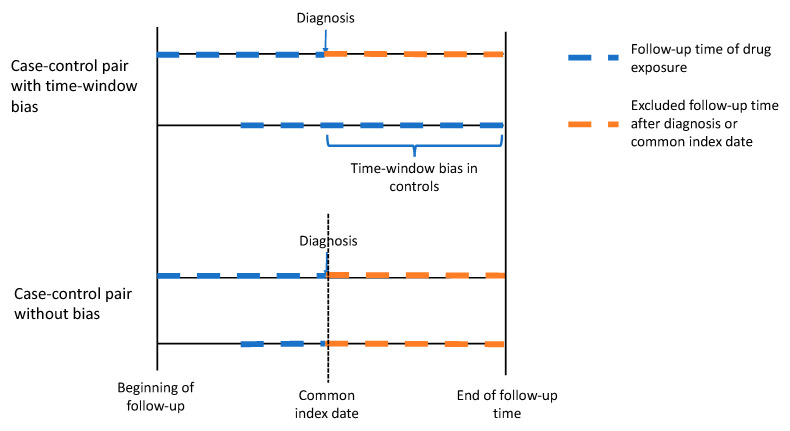
Illustration of time-window bias in an observational pharmacoepidemiological case–control study. This bias occurs when the limitation for drug exposure differs between the cases and controls; exposure is limited to occur before cancer diagnosis among the cases, but no such limitation is applied among the controls. The time-window bias can be avoided by assessing equal time windows for exposure both in cases and their matched controls.

**Figure 3 cancers-13-00696-f003:**
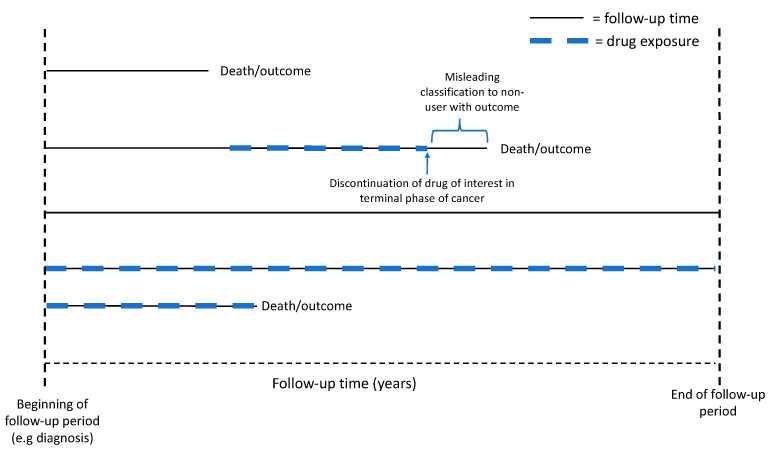
Illustration of bias caused by selective discontinuation of drugs in the terminal phase of cancer and the use of time-dependent variables. This bias may create an illusion that non-users, who might well have been users for most of the follow-up time but discontinue usage in the terminal phase of cancer (user with outcome), are more prone to die. This bias can be avoided by keeping subjects as non-users until the first exposure of drugs, and after that, they remain as ever-users throughout the whole follow-up period.

**Figure 4 cancers-13-00696-f004:**
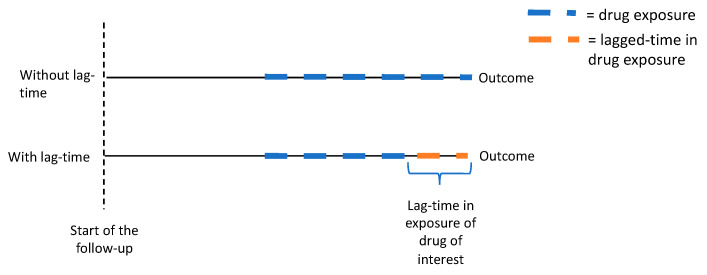
Illustration of lag-time analysis to avoiding protopathic bias. The exposure of drug is lagged in the follow-up time, e.g., with a one-year time lag, the cancer risk is not evaluated by medication use that occurred at time of diagnosis, but by usage that occurred up to one year before that time point.

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
