# Peer review of "Pharmacoepidemiological Evaluation in Prostate Cancer—Common Pitfalls and How to Avoid Them"

_cancers, 2021, doi:10.3390/cancers13040696_

Round 1
Reviewer 1 Report
As the title says, this review is a perspective on the pharmaco-epidemiological study of prostate cancer. The authors are all certainly experts and knowledgeable in pharmaco-epidemiological related topics, and they highlighted several essential points in this work. However, I found not enough connection between the pharmaco-epidemiological section and the prostate cancer section in this review paper. In other words, this manuscript looks like two mini-reviews rather than a single review, while the section related to prostate cancer is not in-depth and comprehensive. This disconnection can be seen again in the discussions and conclusion sections. Therefore, unfortunately, I cannot recommend it to be considered for publication in this format.
Author Response
We understand the point. However, purpose of this manuscript is to describe commonly occurring biases in pharmacoepidemiological research and ways to evaluate and avoid them, especially in prostate cancer research. Many of these issues are universal to cancer research in general, which may give the impression of disconnection mentioned by the reviewer. We have revised the text to highlight and tie these topics especially to context of prostate cancer wherever feasible. All changes are marked using red colored text.
Reviewer 2 Report
The authors address a very important topic and challenge in the field of prostate cancer for conducting pharmacoepidemiological research, though this topic has relevance to the study of the roles of specific classes of drugs on the development and aggressiveness of multiple cancer types . They highlight critical biases that are often present when conducting pharmacoepidemiological studies in cancer, and they provide cogent examples for how such biases can be avoided. The figures in the article are very beneficial to highlighting and clarifying the meaning of these biases. The implications of this work are farther reaching than just for prostate cancer, though it serves as an excellent model to highlight the biases and solutions discussed.
The paper is well organized and written clearly. The reference list is thorough.
There are a couple of minor typos (both relate to use of term "data" which is plural--I recommend to check the rest of the paper to correct any other errors):
Lines 416-417: "...if the data do not....
Line 432: "If data are available..."
Author Response
We are thankful for the reviewer’s evaluation on our review. The language of our manuscript has been edited by English language editor and all typos are corrected.
Reviewer 3 Report
Siltari and colleagues describe common ways pharmacoepidemiological studies might have systemic weakness, and ways that these problems with data analysis might be avoided, especially as applied to prostate cancer. This is a useful topic, especially given the recent interest in ways that drugs used for metabolic or cardiovascular disease might impact prostate cancer. The authors clearly explain how these type of studies might be used in prostate cancer research. They nicely illustrate the types of bias and statistical fashion in ways that could be understood by a general scientific audience.
This work could be improved in a few ways:
- There are scattered English grammatical errors, especially in section 2 (Biological rationale for pharmacoepidemiological studies in prostate cancer). Please examine lines 51, 65, 73, 84, 92, 98, 100, 117, 129, 167, 179, 330, and 401.
- There is no reference for allopurinol on line 135.
- The work uses examples from databases in the US and Finland. It might be useful to mention other potential databases; for example the UK National Health Service.
- There appears to be a misplaced reference on line 109; "Jinx."
Author Response
We are grateful for the reviewer’s feedback from our review.
Response point 1: The language of our manuscript has been edited by English language editor and all grammatic errors are now corrected.
Response point 2: Thank you for carefully reading. We have now added missing references after this sentence.
Response point 3: Example is now added to the text (line 152).
Response point 4: Missing reference is now added.